# Measuring Environmental Resilience Using Q-Methods: A Malaysian Perspective

**Hisham Tariq [1,*]**, **Chaminda Pathirage [2]**, **Terrence Fernando [1]**, **Noralfishah Sulaiman [3]**, **Umber Nazir [3]**, **Siti Kursiah Kamalia Abdul Latib [3]** and **Haidaliza Masram [3]**

1   School of Science, Engineering & Environment, University of Salford, Salford M5 4WT, UK
2   School of Architecture and Built Environment, University of Wolverhampton, Wolverhampton WV1 1LY, UK
3   KANZU Research, Resilient Built Environment (RBE), Universiti Tun Hussein Onn Malaysia, Parit Raja 86400, Malaysia
*   Correspondence: h.tariq4@salford.ac.uk

**Abstract:** Communities increasingly need tools that can help them assess the environmental risks they face to understand better their capacities in mitigation, preparedness, response, and recovery. Environmental resilience (ER) is a crucial feature of community resilience that is not adequately covered in the literature. This paper proposes an inclusive, participatory approach to achieve stakeholder engagement on the definitions, objectives, and indicators for measuring ER at the community level. This study uses a 5-step approach utilising Q-methods to contextualise a resilience index for Environmental Resilience (ER). An initial set of 57 indicators from 13 frameworks from the literature was reduced to 25 by combining the indicators of similar type, format and terminology. A total of 10 participants from two groups (academics and practitioners) took part in the interviews and Q-sort workshops in Malaysia in this study. Both stakeholder groups identified Ecosystem monitoring as one of the most critical indicators to understand ER, closely followed by rapid damage assessments and an effective communication system. The exercise also revealed marked differences between them regarding the importance of fair access to basic needs and services for citizens, a priority for academics, and the value of building green infrastructure, a priority for practitioners, with the most significant difference between the two groups on the importance of measuring the natural defences of a community. The Environmental Resilience Capacity Assessment Tool (ER-CAT), proposed in this paper, can be used by local governments and communities for engagement, discussion and consensus building to select the resilience indicators that are most relevant to them in their contexts.

**Keywords:** resilience indicators; participatory methods; Q-methods; community; disaster management; Environmental Resilience (ER); Malaysia; boundary objects; planning

## 1. Introduction

Across the world, both in developing and developed country contexts, community disaster resilience (CDR) has increasingly been recognised as a vital goal of local, regional, and national governments [1]. The increased uncertainty from climate change impacts has made resilience planning a complex task, with many stakeholders having differing opinions and perspectives on what issues should be addressed and when [2]. Despite the emphasis on analysing resilience issues at the community level, very few resilience frameworks include tools for conceptualising resilience from local perspectives and contextualising its assessment according to the needs of the stakeholders involved in the resilience measurement process [3]. The recent Global Assessment Report (GAR) from the United Nations Office for Disaster Risk Reduction (UNDRR) has highlighted the importance of using a more inclusive, whole-of-society approach to risk assessment and planning [4]. Hence the need for a more participatory approach to resilience assessment where diverse

stakeholder groups can be part of the decision-making process in understanding the risk faced by communities [4].

The concept of CDR, how it is defined and measured, is now increasingly used as a boundary concept to encourage discussion among academics, practitioners, and disaster management professionals [5]. Due to this diversity of stakeholders involved in resilience assessment, there are often differing views on what consists of resilience at the community level and what dimensions can be included within it [6]. Boundary objects can support discourse among stakeholders from diverse fields (i.e., social, economic, health, technological and ecological backgrounds) working towards similar goals but from different perspectives [6]. Several studies have examined the value of using participatory tools as boundary objects for greater stakeholder engagement in sustainability, ecological resilience and economic growth [5–7]. For example, Huggins et al. [5] have shown the value of using resilience as a boundary object to capture the opinions and perspectives of different groups working together in urban planning. With such a range of views in each discipline, multiple definitions of resilience can be proposed and debated resulting in the appreciation of the different approaches to defining and measuring it [6]. In another application, Raadgever et al. [7] use similar boundary objects to set a research agenda for designing interventions for flood management among a diverse group of stakeholders from many different disciplinary backgrounds. Hence, significant debate exists between academics, practitioners, and other community stakeholders on how community disaster resilience and its dimensions are conceptualised, defined, and evaluated, especially for decision-making at different levels [8]. Resilience assessment frameworks need tools to capture some of this debate by allowing for a more subjective approach to understanding the nature and objectives of the different resilience dimensions. Accordingly, planning for community resilience is complex. It requires an interdisciplinary approach that captures the understanding of multiple dimensions of resilience, i.e., the community's physical infrastructure, socio-economic, and environmental resilience (ER) [3].

In Malaysia, recent literature on CDR frameworks has shown the value of using a capital approach to understanding the level of community resilience as a combination of social, economic, and environmental capitals [9] while also highlighting the importance of adapting existing frameworks from other settings to the local context [10]. The Malaysian Government has emphasised resilience planning for ecological hazards as a top priority in legislation such as the National Security Directive 20, which guides local governments and agencies to develop flood Standard Operating Procedures (SoPs) [11]. Creating a long-term sustainable policy for establishing resilient communities at the local levels, with stakeholder consultation, is one of the main objectives of Malaysia's National Disaster Management Authority (NDMA). The NDMA hopes to implement a more inclusive whole-of-society approach to risk assessment and reduction at the state and local levels [12].

The GAR 2022 emphasised increasing environmental risks due to climate change as a significant cause of concern for livelihoods and well-being in developed and developing country contexts in the coming decades [4]. Moallemi et al. [13] have indicated the importance of environmental and ecological resilience in achieving the 17 Sustainable Development Goals (SDGs) targets by including ER indicators in the overall assessment of a community's resilience as a whole system. Despite this importance, a recent survey of CDR literature by Tariq et al. [3] showed that few frameworks explicitly include environmental resilience indicators in their assessment of overall community resilience. The review also showed that few frameworks allow customisation as an option in their implementation, especially for critical dimensions like environmental and ecological resilience as a part of the overall CDR assessment. Consequently, ER at the local level is an essential component of Community Resilience that is often under-represented in CDR assessment frameworks [3].

In the Malaysian context, Sulaiman et al. [10] highlighted the importance of ER for building resilience at the community level, where redundancy can be built into socio-technical and ecological systems for sustainable human and ecosystem growth and development. Malaysia is a country rich in environmental resources with areas of ecological

biodiversity constantly under threat from climate change impacts, economic growth and rapid urbanisation pushing the urban sprawl into previously forested areas [14]. Therefore, developing context-specific ER assessment tools can add value and benefit those involved in creating resilient communities.

This research looks at the opinions and perspectives of practitioners and academics working closely on ER and disaster risk reduction (DRR) at the local levels in Malaysia. The study uses ER indicators compiled from a Library of Community Resilience indicators developed in a previous study by Tariq et al. [3] in a ranking exercise between the two stakeholder groups to form a community-level capacity assessment tool that can help measure ER at the local. The ER Capacity Assessment Tool (ER-CAT) is proposed to achieve consensus between two diverse groups working on the same ER problems. The research seeks to contribute to the overall work on helping intervention design at the local level.

## 2. Community Disaster Resilience

In the context of the post-2015 Millennium Development Goals, the Sustainable Development Goals, and the second phase of the Hyogo Framework for Action, the United Nations (UN) proposed many initiatives to define and measure resilience through appropriate indicators [15]. However, the complexity of the mechanisms at stake and the high heterogeneity of populations, communities and disasters make defining resilience challenging and its measurement even more so [16]. Despite this difficulty, there has been some consensus on what attributes contribute towards community resilience [17].

Nifa et al. [18] stated that the community's role and how it adapts to being prepared for a disaster are central to the CDR concept. Samsuddin et al. [19] defined CDR as when a community can withstand, absorb, and respond to the shock of calamities while maintaining critical functions. Jones et al. [20] have shown how definitions of CDR have changed over the years to include adaptation and transformation. Mohamad et al. [21] described community resilience as a society's ability to embrace change as a priority in raising its overall standard of living. In a recent review of CDR frameworks, Koliou et al. [17] have indicated that a comprehensive definition must include a systems-of-systems approach that reduces three critical attributes: impacts or consequences, recovery time, and future vulnerabilities. Building on these definitions, Tariq et al. [22] propose an approach where stakeholders adjust those definitions to fit their expectations of CDR, resulting in a consensus based on the context and specific circumstances of the community being assessed. In their approach, resilience is considered more comprehensive than just natural disasters. It covers the capacity of a community or urban system to anticipate, absorb and restore performance by building back better from the effects of disruptions due to shocks and stresses from climate hazards or other disturbances [22]. Furthermore, Tariq et al. [22] propose an adaptable CDR framework that includes six dimensions of community disaster resilience–Physical, Human/Health, Economic, Environmental, Social and Governance, as shown in Figure 1. This paper explores the Environmental Resilience dimension and how it can be defined, measured and evaluated in the Malaysian context.

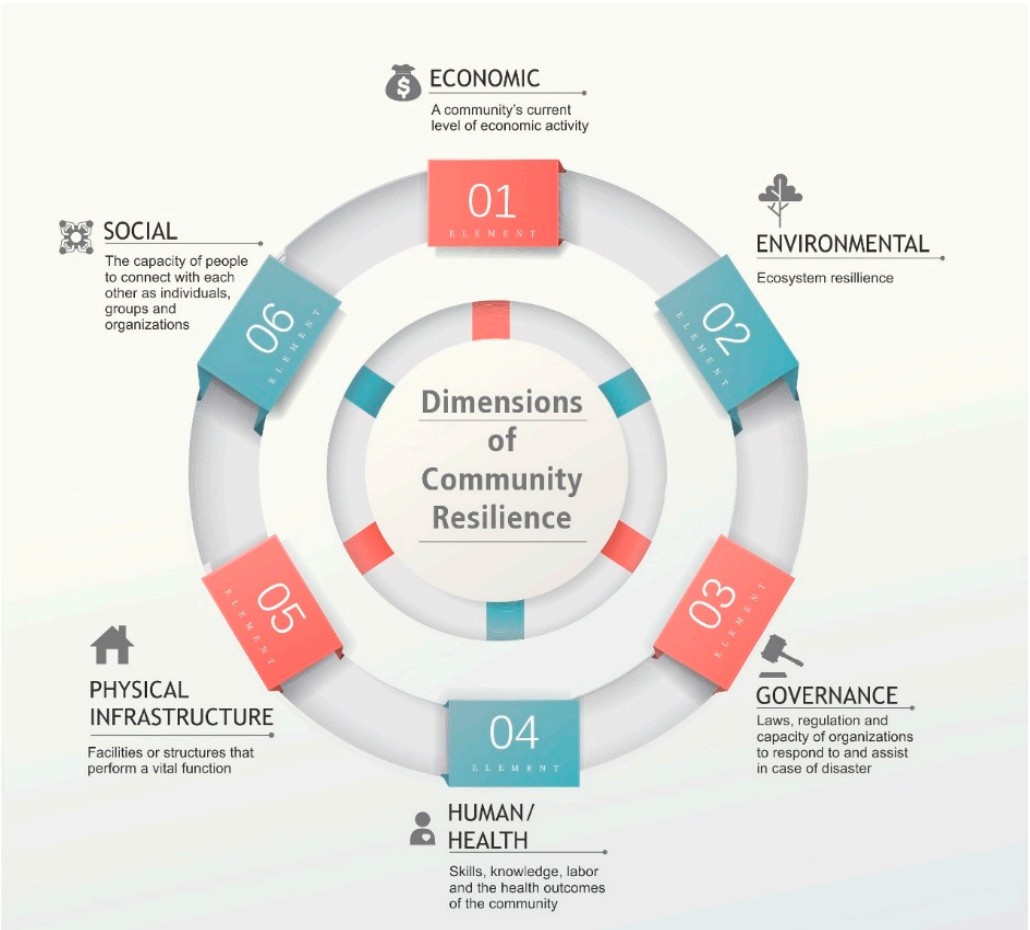

**Figure 1.** Dimensions of Resilience [22].

*Environmental Resilience*

As human societies continue to develop and grow, there are increasing levels of pollution and environmental degradation, resulting in many impacts that can negatively affect livelihoods and well-being [23]. In this research, ecological resilience is considered synonymous with Environmental Resilience (ER), following the definition initially formulated by Holling et al. [24] and expanded upon by Timmerman [25] to link the impact of climate change, environmental hazards and risk mitigation with the overall resilience of the community. Since then, several scholars have tried to quantify it, ranging from a complete assessment of the ecological health surrounding a community [26] to environmental standards scorecards [27] and those based on indexes that measure environmental resilience through indicators [28].

In the Malaysian context, floods are the most common type of disaster, with 9% of the land area in Malaysia flood-prone putting around 4.8 million people living in this area at risk [29]. Accordingly, environmental risk assessment and disaster risk reduction at the local levels are integrated into development planning processes by mainstreaming disaster resilience in the district-level local plans as recommended by the National Policy on Climate Change (NPCC) in 2009 [30]. Hence, ER is a critical component of national policy in Malaysia through the National Security Directive 20 (NSD20), a long-term policy seeking to encourage States to develop Standard Operating Procedures (SoPs) for local Government and communities to tackle issues of environmental hazards [11]. Malaysia has three levels of administrative boundaries; the State level, the district level and the local government levels [11]. Legislation like NPCC and NSD20 ensures that district-level planners consider the environmental aspects of disaster resilience, particularly emphasising its impact on the ER of the community [29,30]. Within this policy framework, the assessment of ER at the

local level is a priority to build community-level resilience to hazards like riverine, pluvial and coastal flooding [12]. To understand resilience in the local context, researchers and academics in Malaysia must work closely with practitioners and stakeholders from the State government and local agencies for better outcomes [14].

Despite being an essential dimension of CDR, Tariq et al. [3] have shown that ER is not as well covered among CDR frameworks as other dimensions like economic or livelihood resilience, physical infrastructure, and social resilience. Of the 36 frameworks in their review, only 13 (36%) included indicators that directly measured ER [3]. This lack of representation among conventional CDR frameworks means that a vital component of a community's resilience is usually missed and thus may lead to a deficiency in any assessment process undertaken using those tools.

Among those CDR frameworks that are more comprehensive, there have been some efforts to quantify ER to improve the overall resilience of communities [31,32]. For example, the National Academy of Sciences in the United States conducted workshops among academic and practitioner groups to derive indicators that cover the following topics: (1) Water; (2) Ecosystem Health; (3) Quality of Life, and (4) Protection of natural assets [27]. In a subsequent follow-up study, the National Academy of Sciences expanded this set of ER indicators to include (1) Biodiversity of species, (2) Green spaces and infrastructure, and (3) Wetlands management along with the others mentioned above [33,34]. In a recent review of ER frameworks in Malaysia, Syed Zainal Yussof et al. [14] indicate that ecological or environmental resilience has been influential in designing techniques for enhancing social resilience to environmental transformations and is vital for resilience planning and mitigation. In their research, they have identified the following main component of ER: (1) Environmental resources such as flora & fauna, biodiversity, and natural resources; (2) the Built Environment, namely land use; (3) Climate Condition, data on temperature and rainfall; (4) Natural Disasters, frequency and magnitude of floods, landslides, earthquakes and other hazards, and (5) Environmental Issues, i.e., waste management and industrial pollution.

Despite these examples from the literature, few frameworks have used participatory approaches to define and evaluate ER [3]. Decision makers at the local level, whether local government officials or community leaders, need additional tools to measure local ecosystems' capacity [35]. Such participatory approaches can help raise awareness, clarify divergence or disagreement, and inform about possible consensus among stakeholders on how to go about measuring ER [22]. Table 1 summarises the indicators taken from the extensive review of CDR frameworks in Tariq et al. [3] and also compiles the list of indicators used for measuring ER resilience from the 13 CDR frameworks that included ER as a critical component.

**Table 1.** Statements of Indicators and their Sources.

| No. | Statement (Indicator Set) | Source(s) |
|---|---|---|
| 1 | Ecosystem monitoring | Cutter et al. [36], Prashar et al. [37] Mayunga [38], Yoon et al. [39] |
| 2 | Rangeland management | National Research Council [33], Prashar et al. [37], Orencio and Fujii [40] |
| 3 | Communication systems | Peck and Simonovic [41], Courtney et al. [42] |
| 4 | Community competence | EPA [43], Saja et al. [44] |
| 5 | Learnability and Training | Courtney et al. [42], EPA [43] |
| 6 | Systems' failure indicators | Yoon et al. [39], EPA [43], Romero-Lankao, et al. [45] |
| 7 | Severity of failure: Environmental loss | Prashar et al. [37], Peck and Simonovic [41] |
| 8 | Community processes (plans) | Saja et al. [44], Siebeneck et al. [46] |
| 9 | Water Supply: quality/quantity | National Research Council [27], Cimellaro et al. [32], Mayunga [38] |

**Table 1.** *Cont.*

| No. | Statement (Indicator Set) | Source(s) |
|---|---|---|
| 10 | Community goals (Efficacy) | EPA [43], Saja et al. [44] |
| 11 | Conservation Strategies (soil, forest) | National Research Council [33], Courtney et al. [42] |
| 12 | Biodiversity Index | Cimellaro et al. [32], National Research Council [33] |
| 13 | Fair Access to basic needs and services | National Research Council [27], EPA [43], Saja et al. [44] |
| 14 | Community inclusiveness and equality | Saja et al. [44] |
| 15 | Environmental damage assessments | Courtney et al. [42], UNDP [47] |
| 16 | Green infrastructure | National Research Council [33], Peck and Simonovic [41], UNDP [47] |
| 17 | Local Cultural beliefs and norms | Saja et al. [44], Alshehri et al. [48] |
| 18 | Recovery time | Yoon et al. [39], Peck and Simonovic [41] |
| 19 | Clean up costs | Cimellaro et al. [32], Courtney et al. [42], UNDP [47] |
| 20 | Mental health | EPA [43], Alshehri et al. [48] |
| 21 | Natural defences | Courtney et al. [39], EPA [43], UNDP [47] |
| 22 | Reforestation/green spaces | National Research Council [33], EPA [43], Romero-Lankao, et al. [45] |
| 23 | Knowledge and learning about hazards | Courtney et al. [42], UNDP [47] |
| 24 | Ecosystem protection | National Research Council [27], EPA [43] |
| 25 | Religious beliefs and norms | Saja et al. [44], Alshehri et al. [48] |

Accordingly, a solution to how ER can be operationalised with elements of participation can be helpful in potentially informing decision-making at the local levels [49]. ER assessment tools can then be used as potential "boundary" objects for discussions to understand the priorities and needs of the different stakeholder groups in a community resilience situation [5]. In this research, an approach is developed to address some of the above challenges regarding greater stakeholder engagement through applying subjective, perspective-based approaches for defining and evaluating ER, using measures and indicators from the established resilience measurement literature.

**3. Methodology**

This paper reports on a novel participatory approach for selecting ER measures by the concerned stakeholders. This study considers ER as a critical dimension in measuring CDR. It looks at how different stakeholder groups define ER as a concept before asking about the best measures/indicators suited to measuring it. The Q-methods approach operationalises ER using a participatory tool that allows for the analysis of individual perspectives and uses quantitative factor analysis to identify any patterns of opinion, uniting and opposing, both within and between groups [50]. One of the benefits of using the method is that it does not require forming a prior hypothesis on perspectives in advance; the results show the patterns of opinions as they exist in the individuals in a group [5]. According to Watts and Stenner [50], another strength of the method is that it can be used on a relatively small sample of individuals. However, the results are not intended to be generalised to a broader population.

The Q-method approach utilises a series of statements as the domain of communicability, or the sum of topics, measures and indicators, within a particular context [51]. These statements are sorted by participants, each providing a viewpoint on what they think are the most important statements from their perspective [5]. When taken in aggregate, these perspectives inform which set of statements represents the point of view of a specific stakeholder group—or at least the range of viewpoints within a group. Figure 2 illustrates the steps taken, methods used and the number of participants in the study.

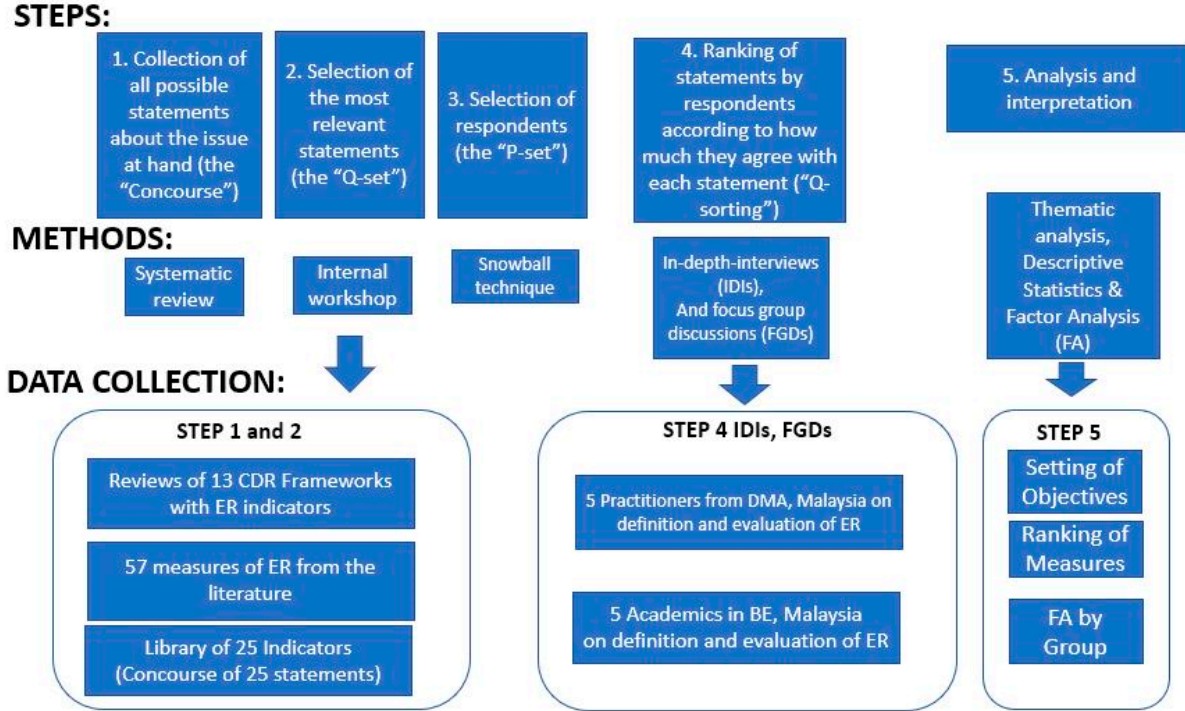

**Figure 2.** Steps of the study, methods used and the number of participants (CDR = Community Disaster Resilience, DMAs = Disaster Management Agencies, ER = Environmental Resilience, BE = Built Environment, FA = Factor Analysis).

Figure 2 shows the five steps undertaken in the Q methods approach used in the study, alongside which methods were used at each stage to understand better the phenomenon under investigation, as indicated by Watts and Stenner [50]. Complex concepts like community resilience and ER are often defined and understood differently by the different stakeholder groups at the local levels in the community. This study's approach uses Q-methods to identify a range of views on measuring ER and then describes these from the perspective of each stakeholder group [52].

In this study, the stakeholders were identified as experts/academics working in the built environment and practitioners working in disaster management agencies. Both groups were working on CDR in a region in Malaysia. The participatory approach developed by Tariq et al. [22] for co-creating Capacity Assessment Tools with different stakeholder groups was adapted here to apply to ER in the two groups identified as Academics and Practitioners. Interviews were used to define ER (and its constituent sub-dimensions), and a Q-sorting exercise was used to construct an index of indicators that can help in the assessment of the ER of a community at the local level. The research team used Q-sorting to allow participants to rank and select the measures according to the Q-sort procedure [53] that will be included in the Environmental Resilience Capacity Assessment Tool (ER-CAT). The composition and breakdown of the participants are shown in Table 2 below.

Participants in the study were selected from academicians within the Resilient Built Environment and practitioners from Disaster Management Agencies (DMAs) that were both working on ER and disaster risk reduction in a regional project. These stakeholders were working to enable local communities to deal with environmental hazards, such as coastal and riverine flooding, and their environmental impacts. Both groups worked on the ER topic from different perspectives and were professionally involved in how ER can be defined, measured and evaluated by their separate organisations.

**Table 2.** Participants' by speciality and experience.

| No. | Speciality | Stakeholder Group | Experience (Years) |
|---|---|---|---|
| 1 | Built Environment | Academic | 15+ |
| 2 | Built Environment | Academic | 7+ |
| 3 | Waste Management | Academic | 5+ |
| 4 | Facilities Management | Academic | 10+ |
| 5 | Construction | Academic | 5 |
| 6 | Disaster Management | Practitioner | 6+ |
| 7 | Disaster Management | Practitioner | 10+ |
| 8 | Crisis Response | Practitioner | 8+ |
| 9 | Crisis Response | Practitioner | 4 |
| 10 | Environmental Agency | Practitioner | 3 |

The approach of engaging stakeholders for the co-creation of the ER-CAT ensures that the assessment is grounded in the perspectives of the concerned stakeholders, making it more relevant, contextual, and designed for use than other approaches [8]. Resilience assessment tools based on greater stakeholder engagement and participatory approaches are more likely to capture local resilience considerations and avoid being influenced by researchers' bias [20]. For instance, Huggins et al. [5] use Q-methods on a similar number of participants from two stakeholder groups in a case study in New Zealand to determine viewpoints for resilience planning. The method is designed for small, selected samples of individuals, and caution must be taken as it is not meant to be generalised to a larger population; hence, its appropriateness for ranking among the expert groups used in this study [53,54].

### 3.1. Developing the "Concourse" or the Sample

This paper focuses on the ER dimension and uses the library of indicators for ER measures as developed in a previous study to explore how resilience is understood in the two stakeholder groups which participated in this study. The review results were used as the "Concourse" in this study, creating an initial library of 57 measures of ER. These measures were then looked at in detail by the research team and were highlighted for further refinement and validation through several steps of the research study to form 25 final statements (representing indicator sets) covering the 57 measures found at the first stage of the study. Other studies, for example, Barry and Proops [55], have also used a similar number of statements to understand stakeholders' perspectives on sustainability and ecological economics. In the subsequent stages, these refined indicator sets were used as statements in the "Q set" (step 2 in Figure 1). Table 1 lists the set of statements and their sources from the literature.

### 3.2. The Procedure

A total of ten respondents, five academics and five practitioners, were interviewed and asked to participate in the Q-sort procedure. Figure 1 and Table 2 both indicate the number and types of participants in the study. In Step 4, stakeholders were interviewed on the definitions of ER and the challenges faced in measuring them. They then participated in a Q-sort exercise that ranked the 25 statements in order of importance to them, thus revealing their preferences on operationalising the concept of ER. For the Q-sorts conducted during the study, the indicators were converted to statements on cards used on the MURAL online collaboration website. Each card represented an indicator set, as shown in Table 1. MURAL allowed the researchers to interact and conduct the interview and the Q-sort exercise with them at a convenient date and time. Participants were asked to define ER in their own words and state the challenges to measuring ER in their contexts. Finally, they were invited to participate in a Q-sort to select the best measures to form an index for the ER-CAT while explaining their choices. Due to the challenge of working in the COVID environment, the interviews, q-sort exercises, and focus group discussions (FGDs) conducted in Step 2 were

only implemented online by the research team. They took about forty-five minutes to one hour each.

The sorting process allowed stakeholders to include essential measures in the ER-CAT and to drop others, ranking them in order of preference from the most important (+4) to the least important (−4). Q-sort uses a forced choice, a quasi-normal sorting distribution designed for use with a 25-item Q-set. The Q-set contains nine ranking values ranging from +4 to −4, which sets the number of items at each value (one at +4, two at +3, and so on). The more Q-sort exercises undertaken with stakeholders, the more extensive the library of available ER-CAT preferences can become. Hence, offering additional insights into how different stakeholders think about ER and their link to the overall community resilience problem being considered. The Q-sorts were entered into Ken-Q Analysis (Version 1.0.7), which was then used to generate descriptive statistics, and the Centroid Factor Analysis (CFA) was conducted in the next stage.

### 3.3. Descriptive Statistics and Centroid Factor Analysis

Applying Q methods allows researchers to explore the main perspectives that might emerge from the combined Q-sort data. Ken-Q Analysis can output the descriptive statistics from the Q-sorts, including the two groups' average rankings separately and then combined [56]. The software was also used to conduct CFA to capture the underlying patterns in the data that can assist in understanding what major perspectives emerge from the combined groups as factors [50]. Ken-Q Analysis uses factor analysis to explain the variance among the participant Q-sorts in terms of the emerging patterns of perspectives. CFA entails analysing the patterns of commonality between the q-sorts by first looking at the correlation or degree of similarity between each Q-sort and then extracting a portion of the common variance explaining the similarity one factor at a time [52]. Each group can be described as a factor revealing their viewpoint on the topic. Using Ken Q-analysis, Varimax rotation is then used to maximise the total variance between the factors. The analysis results in an ideal Q-sort, representing each significant factor and illustrating the perspective that emerges from the whole dataset. These factor-wise ideal Q-sorts can be used to understand what statements (or indicators in this study) are the most important from the perspective of both groups—leading to a potential consensus between them.

## 4. Results

The research team conducted q-sorts and interviews with participants from two groups: academic researchers at a Built Environment research centre and practitioners/professionals from Disaster Management Agencies in Malaysia. The two groups were working together as part of a project on community resilience issues in disaster risk reduction and risk-sensitive urban design in a region of Malaysia. In addition to completing the Q-sorts, they were asked about their views on measuring community resilience and defining and measuring ER from their perspective and case contexts.

### 4.1. Defining Community Resilience from a Malaysian Perspective

The concept of defining and measuring ER separately from community resilience was found challenging by all the respondents. The respondents reflected on this difficulty in separating ER from the other dimensions of resilience. Most respondents ($n = 6/10$, 60%) agreed that ER was an essential aspect of Community Disaster Resilience (CDR), along with several other factors that they considered a priority to better understand and measure the concept, namely: (1) Social Resilience (n = 7/10, 70%) which was highlighted as one of the main elements of CDR; (2) elements that constitute physical resilience (6/10, 60%) such as transportation, utilities and essential services, and (3) livelihoods or economic resilience (6/10, 60%). Other factors that were mentioned included capacities to respond and recover (4/10, 40%), to bounce back and build better (4/10, 40%) and to continue to function in the face of adversity (3/10, 30%).

### 4.2. Challenges Faced in Measuring CDR

When asked about the challenges in measuring CDR, most respondents (8/10, 80%) indicated that data availability was the primary issue facing researchers measuring resilience. From those, several (3/10, 30%) mentioned that Community Level data is not readily available in Malaysia and requires better systems in place to capture. A lack of knowledge and awareness of the public on environmental issues, as well as a lack of funding for research on community resilience, was indicated separately by one respondent each.

### 4.3. Defining Environmental Resilience

Most respondents (8/10, 80%) had difficulty in answering the question on defining ER as a separate concept from CDR and the hazards being faced by the community. When probed, several (3/10, 30%) described it as an essential component or subset of total CDR. Only one managed to offer a clear definition of ER that included elements of ecological resilience, biodiversity and the importance of protection of natural defences at the local level. The difficulty faced by the respondents indicates that it is not easy to define what ER is, that there is a vast diversity of opinions about what it is, and, hence, highlights the problem of measuring and understanding it correctly.

### 4.4. Challenges Faced in Measuring ER

Most participants (7/10, 70%) were hesitant to mention separate challenges and indicated the same ones for measuring overall CDR. Several (4/10, 40%) noted that there is a distinct lack of processes in Malaysia at the local level that can generate the environmental data required for an accurate assessment, which needs more legislation. Among these challenges, several (3/10, 30%) mentioned a lack of knowledge of environmental issues within the public and private sectors. Two participants (2/10, 20%) cited public sector organisations' lack of knowledge regarding environmental laws and regulations. One mentioned the absence of an effective punishment and reward system that can help "nudge" actors to do what is necessary to help monitor and protect local ecosystems.

### 4.5. Q-Sort Exercises

In the Q-sort exercises of Step 4, participants were asked to rank the Q-sets according to their preferences. This set of Q-sort exercises resulted in the finalisation of the ER-CAT, which consists of the twenty-five final indicators. The results from these Q-sorts are summarised in Table 3, where the average ranks of each statement are also shown.

Table 3 shows the overall and the average rankings in the Academic and Practitioner groups separately, including the difference in ranking for contrast. During the interviews, some Academic group participants shared that ER is an aspect of a community's social resilience. This view is reflected in the high ranking given to the indicators on Fair access to basic needs and services, Ecosystem monitoring and protection, and the importance of Communication systems in delivering the message about environmental issues and hazards to the community. In contrast, the Practitioner group indicated that the importance of Green Infrastructure, plans for rapid Environmental damage assessments and the ability to identify and measure Natural defences were critical to understanding ER at the local level.

Overall, with the perspectives of both participant groups combined, it was found that Ecosystem monitoring was the most important set of indicators required for measuring ER. The importance of ecosystem monitoring by both groups is perhaps due to the Malaysian Government including legislation for monitoring tools in its most recent National Disaster Management Plan [30]. Ecosystem monitoring and protection also allow for data generation, which many participants alluded to in the interviews when asked about challenges. Additionally, conducting a rapid and accurate assessment of Environmental damages at the community level was also a key indicator, given its importance in the overall rankings. Environmental damage assessments were critical to the practitioner group as it was part of their everyday processes that they were most involved with (Practitioner 3 interview). Next in importance were the Communication systems, rated by both groups as essential,

without which members of the public cannot be informed and brought into the practice of processing the information and adjusting their behaviour when considering issues of ER (Academic 1, Practitioner 3 and Practitioner 4 interviews). Third and fourth on the list were Fair access to basic needs and services and Green Infrastructure, ranking first among Academics and Practitioners groups, respectively, and still of critical importance overall.

**Table 3.** Capacities and ranking of measures/indicators for ER resilience assessment.

| No. | Environmental Resilience Indicators | Ranking of Measures | | |
|---|---|---|---|---|
| Overall Rank | Statements | Academics | Practitioners | Difference |
| 1 | Ecosystem monitoring tools | 2 | 6 | 4 |
| 2 | Environmental damage assessments | 5 | 2 | 3 |
| 3 | Communication systems | 4 | 5 | 1 |
| 4 | Fair Access to basic needs and services | 1 | 12 | 11 ˆ |
| 5 | Green infrastructure | 12 | 1 | 11 ˆ |
| 6 | Ecosystem protection legislation | 3 | 9 | 6 |
| 7 | Community competence | 6 | 4 | 2 |
| 8 | Water Supply: quality/quantity | 7 | 7 | 0 |
| 8 | Natural defences | 16 | 3 | 13 * |
| 9 | Community goals (Efficacy) | 14 | 21 | 7 ″ |
| 10 | Systems failure indicators | 8 | 10 | 2 |
| 11 | Knowledge and Learning about hazards | 11 | 11 | 0 |
| 12 | Learnability and Training | 13 | 8 | 5 |
| 13 | Biodiversity Index | 9 | 16 | 7 ″ |
| 15 | Reforestation/green spaces | 17 | 15 | 2 |
| 16 | Recovery time | 15 | 14 | 1 |
| 17 | Severity of failure: Environmental loss | 20 | 17 | 3 |
| 18 | Rangeland management | 18 | 18 | 0 |
| 19 | Conservation Strategies | 10 | 13 | 3 |
| 20 | Community inclusiveness and equality | 22 | 19 | 3 |
| 21 | Community processes (plans) | 19 | 22 | 3 |
| 22 | Clean up costs | 21 | 20 | 1 |
| 23 | Local Cultural beliefs and norms | 23 | 23 | 0 |
| 24 | Religious beliefs and norms | 24 | 24 | 0 |
| 25 | Other . . . | 25 | 25 | 0 |

* indicates the largest difference, ˆ indicates 2nd largest difference, and ″ as 3rd largest difference.

On the end of the spectrum, there was a consensus among the respondents (9/10, 90%) that the indicators measuring mental health, religious and cultural beliefs and practices were of little or no practical use where ER issues are concerned. Surprisingly, indicators measuring Community Inclusiveness and Inequality were ranked low in both groups. Some participants (4/10, 40%) indicated that this was not an issue in Malaysia, where all ethnic, religious and social groups were officially considered equal regarding access to resources. Similarly, indicators of a community's ability to make plans were ranked low as most participants (6/10, 60%) considered this as a feature absent from Malaysian society at the local levels–perhaps indicating the lack of legislative or procedural foundations for communities to make and execute their plans at the local levels.

There was a high degree of variability in the indicators' ranking, as reflected in the Q-sorts of both participant groups. Table 3 also shows where the highest level of contrast occurs between the two groups. This contrast is especially apparent in the selection of indicators measuring a community's status or availability of natural defences; the academics ranked it 16th while the practitioners ranked it at 3rd most important. Interestingly, practitioners emphasised the importance of measuring man-made Green Infrastructure projects (ranked 1st) that could dampen the impact of environmental hazards. However, the academics did not view it as necessary (ranked 12th).

Similarly, the academics group underlined the importance of Fair Access to basic needs and services amongst the populace (ranked 1st), and the practitioners gave it a

lesser ranking (ranked 12th) for their measurement of ER. The Table shows that apart from the indicators for measuring Community Goals and the Bio-diversity Index, most other contrasting differences among the indicators were not overly pronounced.

In addition to the ranking of measures, Table 4 shows the degree of similarity in the Q-sorts of each participant. The more similar the rankings in the Q-sort, the closer the correlation is to 100. The Table shows a high degree of spread among the academics and the practitioner groups, indicating the diversity of views on measuring ER within and between the two groups. The next set of statistical analyses, presented in Table 5, is based on the variability between the Q-sorts and will help us explore the data further.

**Table 4.** Correlation between the individual Q-sorts.

| Respondent | Academic Sort 1 | Academic Sort 1 | Academic Sort 1 | Academic Sort 1 | Academic Sort 1 | Practitioner Sort 1 | Practitioner Sort 1 | Practitioner Sort 1 | Practitioner Sort 1 | Practitioner Sort 1 |
|---|---|---|---|---|---|---|---|---|---|---|
| Academic Sort 1 | 100 | 20 | 35 | 23 | 59 | 35 | 8 | 41 | 23 | 3 |
| Academic Sort 2 | 20 | 100 | 3 | 20 | 45 | 48 | 17 | 38 | 44 | 18 |
| Academic Sort 3 | 35 | 3 | 100 | 51 | 20 | 49 | 55 | 26 | 47 | 48 |
| Academic Sort 4 | 23 | 20 | 51 | 100 | 27 | 54 | 70 | 37 | 39 | 64 |
| Academic Sort 5 | 59 | 45 | 20 | 27 | 100 | 33 | 17 | 51 | 33 | 31 |
| Practitioner Sort 1 | 35 | 48 | 49 | 54 | 33 | 100 | 33 | 32 | 38 | 27 |
| Practitioner Sort 2 | 8 | 17 | 55 | 70 | 17 | 33 | 100 | 12 | 29 | 73 |
| Practitioner Sort 3 | 41 | 38 | 26 | 37 | 51 | 32 | 12 | 100 | 64 | 25 |
| Practitioner Sort 4 | 23 | 44 | 47 | 39 | 33 | 38 | 29 | 64 | 100 | 41 |
| Practitioner Sort 5 | 3 | 18 | 48 | 64 | 31 | 27 | 73 | 25 | 41 | 100 |

**Table 5.** Covariance between rotated factors.

| No. | Participant | Factor 1 | Factor 2 | Factor 3 |
|---|---|---|---|---|
| 1 | Academic Sort 1 | 0.4463 | −0.3028 | 0.0956 |
| 2 | Academic Sort 2 | 0.4581 | −0.4124 | 0.1818 |
| 3 | Academic Sort 3 | 0.6235 | 0.2205 | 0.0265 |
| 4 | Academic Sort 4 | 0.7337 | 0.3522 | 0.0835 |
| 5 | Academic Sort 5 | 0.5858 | −0.3769 | 0.1496 |
| 6 | Practitioner Sort 1 | 0.6553 | −0.0269 | 0.0021 |
| 7 | Practitioner Sort 2 | 0.5816 | 0.6302 | 0.4028 |
| 8 | Practitioner Sort 3 | 0.6066 | −0.3796 | 0.1518 |
| 9 | Practitioner Sort 4 | 0.6747 | −0.1029 | 0.0142 |
| 10 | Practitioner Sort 5 | 0.615 | 0.469 | 0.1687 |
| | | **Factor 1** | **Factor 2** | **Factor 3** |
| | Eigenvalues | 3.6484 | 1.349 | 0.2862 |
| | % Explained Variance | 36 | 13 | 3 |
| | Cumulative % Explained Variance | 36 | 49 | 52 |

In applying Q-sorts for analysis, Watts and Stenner [50] recommend conducting a centroid factor analysis that can reveal the ideal Q-sort, called factor array, for each factor found significant. This study extracted three centroid factors from the Q-sort data set, which were input into the Ken Q-analysis software. Two factors were found to be statistically significant using the Kaiser-Guttman criterion of Eigenvalues greater than one, as indicated by Zabala and Pascual [57] and shown in Table 6 below. Watts and Stenner [50] recommend using Cattell's Scree Plot diagram to decide the number of factors where the line changes slope, as shown in Figure 3 below. They also suggest a rule of thumb of having at least one factor for every 5 to 6 participants. For the CFA analysis, Ken Q-analysis software was used to conduct factor rotations to observe the factor loadings between the first and the second factors and tested it between the first and third factors, but it was found not statistically significant. Ken Q-analysis allowed the data to be checked to seven factors as recommended by Brown and Rhoades [54], with both the Kaiser criterion and the Scree test indicating that three factors were sufficient in this case [54]. Table 5 shows that Factor 1 explains 36 per cent of the variance in the Q-sorts, while Factors 1 and 2 together can account for 49 per cent of the cumulative variance.

**Table 6.** Factor A and B Ideal Q-sorts.

| Statement Number | Statement | Factor 1 | Factor 2 |
|---|---|---|---|
| | | Sort Values | Sort Values |
| 1 | Ecosystem monitoring | 1 | 3 |
| 2 | Rangeland management | −2 | 1 |
| 3 | Communication systems | 3 | −2 |
| 4 | Community competence | 4 | −1 |
| 5 | Learnability and Training | 1 | −1 |
| 6 | Systems failure indicators | −1 | 0 |
| 7 | Severity of failure: Environmental loss | −1 | 1 |
| 8 | Community processes (plans) | 0 | −2 |
| 9 | Water Supply: quality/quantity | 2 | 0 |
| 10 | Community goals (Efficacy) | 0 | 0 |
| 11 | Conservation Strategies | −2 | 4 |
| 12 | Biodiversity Index | 0 | 1 |
| 13 | Fair Access to basic needs and services | 1 | 0 |
| 14 | Community inclusiveness and equality | −1 | 0 |
| 15 | Environmental damage assessments | −1 | 2 |
| 16 | Green infrastructure | 2 | 3 |
| 17 | Local Cultural beliefs and norms | −3 | −3 |
| 18 | Recovery time | 0 | −1 |
| 19 | Clean up costs | 0 | −2 |
| 20 | Others . . . | −4 | −4 |
| 21 | Natural defences | 2 | 2 |
| 22 | Reforestation/green spaces | −2 | 1 |
| 23 | Knowledge and Learning about hazards | 3 | −1 |
| 24 | Ecosystem protection | 1 | 2 |
| 25 | Religious beliefs and norms | −3 | −3 |

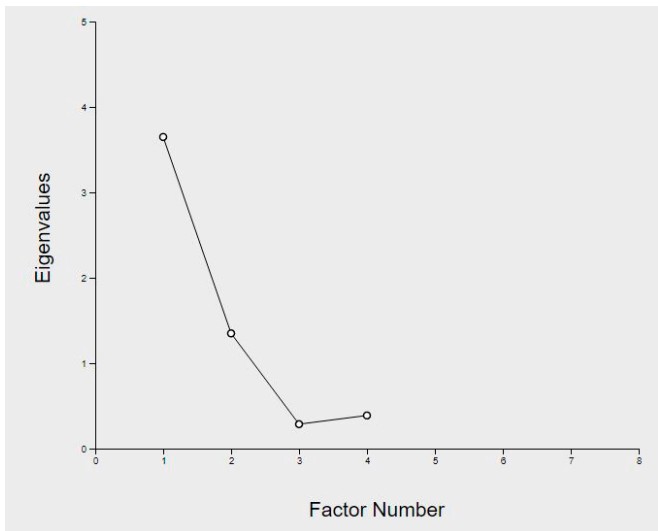

**Figure 3.** Scree Plot.

Finally, the Varimax rotation method was used to derive the ideal Q-sort around the majority viewpoint in both sets of Q-sorts. In the Varimax rotation method, the factors identified in the previous step represent a perspective that can be shown in an ideal Q-sort that forms the prevalent opinion on the discourse around measuring ER in both groups. The ideal Q-sorts are shown below in Table 6. The ideal Q-sort looks at the consensus developed around the two factors found to be statistically significant in the analysis. Factor A can be described as forming a consensus around the importance of communities' capacities, their ability to learn from events and how learning can be internalised and communicated across

the community system. Factor B comprises a consensus around the importance of having an enabling legislative environment where conservation strategies, ecosystem monitoring and green infrastructure are promoted and brought into place. Including these measures in our community-level environmental assessment may lead to greater acceptance by both groups.

## 5. Discussion

The research study sought to understand the prevalent opinions among two groups in building resilience at the local levels. It found that there was indeed a difference in the way both groups envisioned what ER was and how ER should be measured. Several recent reviews of the literature on resilience have shown little consensus on operationalising community resilience [3,17]. Those that have proposed a framework for assessment have underrepresented ER as a core dimension in its measurement [3]. This study sought to address a gap in CDR field studies, as most do not cover ER within their evaluations. The research tries to capture the critical elements of ER and what indicators should be used to measure it from the perspectives of two stakeholder groups working on community resilience issues in a region of Malaysia. More specifically, this research seeks to develop a tool to identify the prevalent opinion patterns among academics working in community resilience and compare them with practitioners working on Disaster Management at the local levels in Malaysia.

Several resilience scholars have recommended including more "subjective" approaches to measuring resilience to allow stakeholder groups' diverse opinions, perspectives, and experiences to be considered in the resilience assessment process [20]. For example, Koppiahraj et al. [58] have used a fuzzy logic approach to rank preferences for sustainability assessment methods used by different practitioner groups. The ER Capacity Assessment Tool proposed in this study emphasises the stakeholders' role in defining what ER means to them and how it can be measured. The approach used in this paper reflects this inclusive philosophy by using a more subjective approach and allowing for the customisation of the assessment according to stakeholder input and the case context.

The adaptable CDR framework developed by Tariq et al. [3] outlines how such an assessment can be tailor-made to stakeholders' requirements and allow for comparison and consensus building among different viewpoints and perspectives. In a broader sense, the approach combines a top-down and a bottom-up approach to propose a middle-out approach that develops a library of indicators taken from a literature review (the top-down part of the exercise) and presents them to participants to arrange according to preference (the bottom-up aspect of the exercise). Hence, each participant prioritises what ER components they want to measure and what indicators to use to measure them.

Table 4 indicates that the two groups have ranked the statements according to their essential criteria. For example, the academic group in this study organised Fair access to basic needs and services, Ecosystem monitoring and Ecosystem protection legislation as the critical indicators. In contrast, the practitioner group ranked Green infrastructure, Environmental damage assessments, and natural defences as the most important in measuring ER at the local levels in Malaysia. It is interesting to compare and contrast this with two well-known CDR frameworks that use different criteria to measure ER. For instance, the PEOPLES framework defines ER with indicators such as (1) Water quality/quantity, (2) Air quality, and (3) Soil quality [32]; and Yi and Jackson [34] looked at quantifying ER across three categories; (1) Forest/Vegetation resilience, (2) soil microbial resilience, and (3) hydrological resilience based on climate data reflecting perhaps the focus for these two groups in Malaysia was different than the authors of those frameworks.

Table 4 also shows that the two groups have some marked differences, especially concerning five indicators: the role of Natural defences, Green infrastructure; Fair access to basic needs and services; Community goals and efficacy; and the Bio-diversity index. As discussed above, the viewpoints of both groups have differed on how vital certain "engineering" or "hard" solutions (such as the availability of Green Infrastructure projects and the presence or absence of Natural defences in the community) are for measuring

community resilience from the practitioners' perspectives. Conversely, from the perspective of the academic researchers, the importance of community-level social indicators like Fair access to basic needs and services and Community efficacy in setting and achieving their stated goals is more significant in their view requiring a priority inclusion for ER measurement from their perspectives.

Table 7 summarises the overall ranking of both groups combined. Again, it is interesting to note that the top-ranking indicators in Table 7 also show up as an outcome of workshops with a mixed group of academics and practitioners held by the National Academy of Sciences in 2015 and 2017 in the United States [27,33]. Although the output of the NRC studies did not rank the indicators by importance, it was based on consensus between the groups. This study validates some of those findings regarding the ER measures proposed in that study in our setting and adds ranking preference between the two groups for more clarity.

**Table 7.** Final Table of Indicators for the ER-CAT.

| Overall Rank | Environmental Resilience Indicators | Example of Indicator/Measure |
|:---:|:---:|:---:|
| 1 | Ecosystem monitoring tools | Are Monitoring Systems in place and working? Does it cover all hazard types? Providing regular and up-to-date data on local natural resources/biodiversity? |
| 2 | Environmental damage assessments (EDA) | Laws/Regulations for conducting EDA in place? How quickly can EDAs be undertaken? Staff/organisations trained for EDA? Cost of damaged ER assets |
| 3 | Communication systems | Early warning system, the mechanism for delivery of environmental advisories, the effectiveness of messaging, number of cell phones/household |
| 4 | Fair Access to basic needs and services | Fair access for all population groups to water, natural resources, and green spaces and other environmental services |
| 5 | Green infrastructure | Availability of Green Infrastructure plans? Approved projects? Funding? Maintenance? |
| 6 | Ecosystem protection legislation | Laws/regulations for the protection of natural resources/biodiversity, System of incentives in place (taxes/subsidies), Punishment |
| 7 | Community competence | Engagement level in policy and planning, social mobilisation levels for ER, education level |
| 8 | Water Supply: quality/quantity | Water supply system functionality, % households with access to clean water, additional fresh water supply sources, stocks, water quality over time |
| 8 | Natural defences | Identified and protected buffer zones, maintained and upkeep of areas, and enforcement mechanisms |
| 9 | Community goals (Efficacy) | Community perception/awareness of ER goals, priorities, and plans to set and achieve ER targets/goals |
| 10 | Systems' failure indicators | Availability and accessibility of natural resources (pre, during and after the event), access to livelihood functions, |
| 11 | Knowledge and Learning about hazards | Awareness, Knowledge of local risks or perceptions, adaptability and learning from past events |
| 12 | Learnability and Training | Number of people trained for DRR and first aid, Number of training programmes initiated in the local area, Exercises and Drills conducted |
| 13 | Biodiversity Index | Biodiversity count (flora/fauna), Species health (flora/fauna) |
| 15 | Reforestation/green spaces | Extant of natural tree cover, Impact of Green spaces initiatives and reforestation activities |

**Table 7.** *Cont.*

| Overall Rank | Environmental Resilience Indicators | Example of Indicator/Measure |
|:---:|:---:|:---:|
| 16 | Recovery time | Time to Clean up pre-event levels |
| 17 | Severity of failure: Environmental loss | Deforestation/erosion protection costs, loss for certain hazard levels (thresholds for different hazards) |
| 18 | Rangeland management | Up-to-date and effective legislation on the Protection level of wetlands and watersheds, Rangeland zoning and enforcement |
| 19 | Conservation Strategies | Impact of wildlife conservation, number and type of conservation programmes, environmental advisories, and initiatives implemented |
| 20 | Community inclusiveness and equality | Gender, Ethnic and special needs population equality and involvement in ER policy |
| 21 | Community processes (plans) | Participation in Planning (community plans), a mechanism for sharing plans across the community, frequency of community meetings |
| 22 | Clean up costs | Clean-up/restoration costs, Timely availability of funds, the mechanism to prioritise funding for clean-up and restoration |
| 23 | Local Cultural beliefs and norms | Cultural and historical preservation of natural resources, Existing cultural and behavioural norms |
| 24 | Religious beliefs and norms | Current religious practice and world view on environmental protection and conservation, Faith-based engagement activities for ER |
| 25 | Other . . . | |

Finally, the Q-sorts generated around the two factors, found to be statistically significant by the Varimax method, also add some additional insight into what indicators we should consider including in the final assessment tool. Among both groups, there is agreement on capturing community capacities, understanding a community's initial ability to prepare for and cope with environmental impacts, and learning from and then communicating those findings. The second set of indicators that gained consensus was around the legislative and governance system that leads to an enabling environment for ecosystem monitoring (generating data for ER as mentioned in the interviews), conservation strategies by the communities themselves, and the enforcement of green infrastructure projects where required.

The ER assessment tool developed through this approach among the two groups in this research can act as a Boundary Object that can help to promote consistent interpretations and responsive information sharing between emergency management collaborators and their different activity systems or a combination of activities [59]. Huggins et al. [5] have indicated that distinct organisational histories and norms of operating can result in many viewpoints. Cop et al. [60] also demonstrate the importance of capturing diverse views when working collaboratively for environmental sustainability. Hence, there is a need to consider the differences and commonalities of each collaborating group to facilitate productive collaborations between the activity systems of diverse teams, boundary objects and associated processes.

Owen et al. [59] suggested that "highlighting the historical and contradictory elements within an activity system . . . enables an analysis of those contradictions to be used to trace disruptions and to point to new opportunities for development." Using Q-methodology allows researchers to find points of contradiction in opinions and perspectives that can lead to consensus or division among two groups working for the same objective. The Q-methods approach is used in this research to tap into current debate and discussion on ER in the two organisations working in Malaysia on Disaster Management and resilience. This research has tried to identify the general opinion patterns about ER indicators by

Malaysian academics in a resilience research group at a local university and by Malaysian practitioners working in disaster management. Additionally, the research has looked at the patterns of opinions indicating agreement between both groups. The approach may be used for developing ER assessment tools to act as a functional boundary object between these groups to help build frameworks, policies and interventions that can contribute to ER at the local levels in Malaysia.

By considering these factors and combining the views of both groups, we can achieve a consensus on how ER can be measured, highlighting the crucial indicators from both groups' perspectives that are essential to the ER assessment process. The ER-CAT developed here can be used to capture the ER of a community at the local levels at a point in time. It can then be used to create a larger overall model of CDR in an adaptable resilience framework that includes other dimensions such as physical, economic, human/health and social aspects.

### 5.1. Limitations

The number of participants used in the study was relatively small. It cannot be extrapolated to represent Malaysia's wider academic research community or the broader disaster management practitioners except in the context of the region used in this study. Due to the health and safety guidelines for conducting research during the COVID pandemic, the two groups were also not brought back together in a joint workshop for a validation exercise on the results and to form a consensus on measuring ER. Additional fieldwork could be undertaken in a subsequent project activity per the pre-requisite health and safety data collection requirements under appropriate COVID research ethics guidelines.

### 5.2. Policy Implications

Although the sample size is small, the tools developed in this study may be used in larger contexts. Q-sorts allow researchers to understand the prevailing ideas and thoughts circulating in stakeholder groups and may be used to capture differing (as well as converging) viewpoints that can lead to consensus building to achieve resilience action in communities. It is proposed that adding these tools to a digital platform may allow for a more user-friendly experience in visualising these preferences and a better understanding of resilience assessment preferences among stakeholder groups and for a greater level of engagement than other tools allow.

### 6. Conclusions

Communities consist of diverse stakeholders with often diverging views regarding resilience issues. Academic researchers on resilience and disaster management practitioners frequently disagree on what community resilience means and how it should be measured. Using a Q-methods as a subjective approach to indicator selection for assessing the ER of a community can help researchers understand the prevailing opinion and perspective on ER measurement among different participant groups. The Q-sorts have allowed the participants to reveal their preferences based on their choices. The research has shown that although two distinct perspectives exist among the participant groups, there is a lot of common ground that can form the basis of a consensus. Using ranking tools like the Q-sort as a boundary object helped inform us about the debates within and between groups that can divide and unite stakeholders that were working together on resilience issues at the local level in one region of Malaysia. The study results can potentially be used for a consensus, or at least an appreciation of other groups' perspectives, for measuring ER.

The ER-CAT introduced in this study is just one outcome of an assessment tool resulting from applying the adaptable CDR framework in the context of ER among two stakeholder groups working on resilience issues at the local levels in Malaysia. The tool can be customised for use in any of the six dimensions of the adaptable CDR framework (Economic, Environmental, Social, Health, Governance and Physical Infrastructure). The Capacity Assessment tool can effectively gain insight into the different stakeholder groups' diverse perspectives, goals, and motivations in a resilience case context. It can help bridge

the understanding between these groups by offering an opportunity to stimulate discussion and, at least, encourage a consensus among those groups.

*Future Works*

A participatory approach to resilience assessments aims for greater inclusivity and a more equitable representation among community groups with a stake in the ER issues of the community. Furthermore, other stakeholder groups can be added to the study to compare and contrast the views of different groups within the community context. People from diverse socio-economic and ethnic, cultural, and religious groups can be included to see their perspectives on the ER concept as it applies to them. The tool could be specifically adapted and designed to account for those marginal groups frequently left out of the consultative process. Additionally, the ER-CAT developed in this research will be applied in a case study of a community to create a simulation model of ER at the local levels. The model could be used to test different scenarios regarding possible interventions and what impacts those interventions can have on the community if applied. The simulation modelling process will enable the academics and practitioner groups to compare and contrast different interventions and consider their potential benefits on the community system as a whole.

**Author Contributions:** Conceptualisation. H.T. and C.P.; methodology, H.T.; Validation, H.M. and N.S.; formal analysis, H.T. and C.P.; investigation, data collection, H.T., U.N. and S.K.K.A.L.; resources, N.S. and T.F.; data curation, H.M., U.N. and S.K.K.A.L.; writing—original draft preparation, H.T.; writing—review and editing, C.P., N.S. and T.F.; visualisation, U.N. and S.K.K.A.L.; supervision, C.P. and N.S.; project administration, U.N. and H.M.; funding acquisition, C.P. and T.F. All authors have read and agreed to the published version of the manuscript.

**Funding:** The authors express their gratitude to the Global Challenges Research Fund (GCRF) and the Engineering and Physical Sciences Research Council (EPSRC) for the financial support under the International Grant, EP/PO28543/1, entitled "A Collaborative Multi-Agency Platform for Building Resilient Communities" for the work reported in this paper. This work was also supported by the Economic and Social Research Council (ESRC) under the Grant ES/T003219/1 entitled "Technology Enhanced Stakeholder Collaboration for Supporting Risk-Sensitive Sustainable Urban Development".

**Institutional Review Board Statement:** The study was conducted under the University of Salford's ethics code requirements and the UK Research Registry Office regulations for studies involving humans and was approved by the Science and Technology Research Ethics Panel of the University of Salford for "A Collaborative Multi-Agency Platform for Building Resilient Communities" project with application ID STR718-59 and date of approval: 31 August 2018.

**Informed Consent Statement:** Informed consent was obtained from all subjects involved in the study.

**Data Availability Statement:** The data presented in this study are available on request from the corresponding author. The qualitative data are not publicly available due to the privacy of individuals who participated in the study.

**Acknowledgments:** The authors wish to thank Simon Hadfield and Hanneke van Dijk for providing valuable support during the research process.

**Conflicts of Interest:** The authors declare no conflict of interest.

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
