# Peer review of "Measuring Environmental Resilience Using Q-Methods: A Malaysian Perspective"

_sustainability, doi:10.3390/su142214749_

Round 1

Reviewer 1 Report

This is a well-written paper that reflects sound research.

Lines 152-154 mentions a 'district local plan'. A bit of context will be helpful here as the readers may not be familiar with the governance structure in Malaysia.

Table 1 has a title which includes "to review before submission"?

Methodology - rather than the 'paper using the Q-method...', I assume it is the research that used the Q-method, and the paper is reporting the research?

A little rationale for using the Q-method will be helpful, over and above other available methods.

The findings are well explained.

It will be good to identify some of the key findings in the abstract, it doesn't shed enough light on what has been discussed in the paper.

In places where author names are mentioned within sentences, you have said like "as initially considered by 24 and expanded upon by 25, where", but it will be better to indicate the author names in situations where the names are needed to make the sentences work and then include the citation number. 

Reviewer 2 Report

Manuscript Title: Measuring Environmental Resilience using Q-Methods: A Malaysian Perspective

Manuscript ID: Sustainability-1989039

The following are my comments on the manuscript:

1.      Abstract: Purpose of the study needs to be more emphasized. On what basis, the indicators were reduced to 25 from 57. Keywords need to be improved.      

2.      Introduction: Misses coherence. The significance of the study needs to be explicitly highlighted and summarized by mentioning the earlier works. Also, the research gaps identified in the earlier literature have to be mentioned. Finally, how this study goes to bridge the identified research gaps have to be mentioned. In the present form, the manuscript fails to provide the above needed information.          

3.      Community Disaster Resilience: This section can be narrowed down. Currently, this section looks vague. Try to keep all the references discussed in this section close to the topic. If the Figure 1 is adopted from earlier literature, reference need to be provided. What is CDRF?        

4.      Methodology: Give some examples where q-method has been used earlier. Are 10 experts sufficient? Explain about Centroid Factor Analysis (CFA).  

5.      Results and discussion: Looks lengthy.   

6.      Discussion: Compare and contrast with earlier studies.

7.      Conclusion: Comprehensively conclude the study by highlighting the contributions of the study. Also, mention the future scope of the study.   

8.      References: In text references are not in correct format.

The following articles may help in better improving the quality of the manuscript:

Çop, S., Olorunsola, V. O., & Alola, U. V. (2021). Achieving environmental sustainability through green transformational leadership policy: Can green team resilience help?. Business Strategy and the Environment30(1), 671-682.

Thomas, E., Bradshaw, A., Mugabo, L., MacDonald, L., Brooks, W., Dickinson, K., & Donovan, K. (2021). Engineering environmental resilience: A matched cohort study of the community benefits of trailbridges in rural Rwanda. Science of The Total Environment771, 145275.

Koppiahraj, K., Bathrinath, S., Venkatesh, V. G., Mani, V., & Shi, Y. (2021). Optimal sustainability assessment method selection: a practitioner perspective. Annals of Operations Research, 1-34.

The paper has focused on one of the pressing issue. However, it has many drawbacks. There are typo errors and language of the manuscript needs attention. Is Appendix A necessary? Hence, it was suggested for major revision.              

Round 2

Reviewer 2 Report

Earlier comments have been addressed. Check the referencing style. Now, the manuscirpt can be considered for publication.